# Depth separation and weight-width trade-offs for sigmoidal neural networks

**Amit Deshpande & Navin Goyal**
Microsoft Research
{amitdesh,navingo}@microsoft.com

**Sushrut Karmalkar**
University of Texas at Austin
s.sushrut@gmail.com

## Abstract

Recent work has shown strong separation between the expressive power of depth-2 and depth-3 neural networks. These separation results exhibit a function and an input distributions, so that the function is well-approximable in $L_2$-norm on the input distribution by a depth-3 neural network of polynomial size but any depth-2 neural network that well-approximates it requires exponential size. A limitations of these results is that they work only for certain careful choices of functions and input distributions that are arguably not natural enough.

We provide a simple proof of $L_2$-norm separation between the expressive power of depth-2 and depth-3 sigmoidal neural networks for a large class of input distributions, assuming their weights are polynomially bounded. Our proof is simpler than previous results, uses known low-degree multivariate polynomial approximations to neural networks, and gives the first depth-2-vs-depth-3 separation that works for a large class of input distributions.

## 1 Introduction

Understanding the remarkable success of deep neural networks in many domains is an important problem at present, see, e.g., LeCun et al. (2015). This problem has many facets such as understanding generalization, expressive power, optimization algorithms in deep learning. In this paper, we focus on the question of understanding the expressive power of neural networks. In other words, we study what functions can and cannot be represented and approximated by neural networks of bounded size, depth, width and weights.

The early results on the expressive power of neural networks showed that the depth-2 neural networks are universal approximators; that is, for various choices of activation functions or neurons, depth-2 neural networks are powerful enough to uniformly approximate arbitrary continuous functions on bounded domains in $\mathbb{R}^d$, e.g., Cybenko (1989); Hornik et al. (1989); Barron (1994). However, the bounds they provide on the size or width of these neural networks are quite general, and therefore, weak. The study of expressive power of neural networks requires understanding what functions can be represented or well-approximated by neural networks with bounded parameters, e.g., the number of neurons, the width of hidden layers, the depth, and the magnitude of its weights.

Natural signals (images, speech etc.) tend to be representable as compositional hierarchies, as argued in LeCun et al. (2015), and deeper networks can be thought of as representing deeper hierarchies. The power of depth has been a subject of investigation in deep learning, e.g., He et al. (2016). To understanding the effect of depth on the expressive power, one may ask whether having more depth allows representation of more functions by a network of bounded size or width.

Eldan & Shamir (2016) show a separation between depth-2 and deptFh-3 neural networks. More precisely, they exhibit a function $g : \mathbb{R}^d \to \mathbb{R}$ and a probability distribution $\mu$ on $\mathbb{R}^d$ such that $g$ is bounded and supported on a ball of radius $O(\sqrt{d})$ and expressible by a depth-3 network of size polynomially bounded in $d$. But any depth-2 network approximating $g$ in $L_2$-norm (or squared error) within a small constant under the distribution $\mu$ must be of size exponentially large in $d$. Their separation works for all reasonable activation functions including ReLUs (Rectified Linear Units) and sigmoids. The input distribution they use is the square of the Fourier transform of the indicator function of the unit ball in $\mathbb{R}^d$, and their proof crucially relies on this.

Safran & Shamir (2017) extend the result of Eldan & Shamir (2016) to more natural functions such as the indicator functions of balls and ellipses, non-linear functions that are radial in $L_1$-norm, and smooth non-linear functions etc. However, their proof technique requires the same underlying input distribution as in Eldan & Shamir (2016).

Daniely (2017) (see also Martens et al. (2013)) gives a separation between depth-2 and depth-3 networks by exhibiting a function $g : \mathbb{S}^{d-1} \times \mathbb{S}^{d-1} \to R$ which can be well-approximated by a depth-3 ReLU neural network of polynomially bounded size and weights but cannot be approximated by any depth-2 (sigmoid, ReLU or more general) neural network of polynomial size with (exponentially) bounded weights. This separation holds under uniform distribution on $\mathbb{S}^{d-1} \times \mathbb{S}^{d-1}$, which is more natural than the previous distributions. However, the proof technique crucially uses harmonic analysis on the unit sphere, and does not seem robust enough to be applicable to other distributions.

Further discussion of previous work appears in Appendix.

Most of the known (and all of the above) separation results fit the following template: certain carefully constructed functions can be well approximated by deep networks, but are hard to approximate by shallow networks using a notion of error that uses a carefully defined distribution. Because of this limitation these results seem to not shed much light on the real life applications. Neural nets have diverse applications to diverse types of data ranging from perceptual to completely synthetic such as those arising in learning to play board games using deep reinforcement learning. To our knowledge, there is no well accepted general theory of data distribution. In this light, for results about expressive power, it becomes important to understand the extent to which they are robust to change in the input distribution. Such an understanding would be a step towards results that are relevant to practice. In particular, we would like to understand whether there are large classes of functions and distributions that witness the separation between deep and shallow networks.

## 2 OUR RESULTS

We exhibit a simple function (derived from Daniely (2017)) over the unit ball $\mathbb{B}^d$ in $d$-dimensions that can be well-approximated by a depth-3 sigmoidal neural network of width and weights polynomially bounded in $d$. However, any reasonable approximation of this function using a depth-2 sigmoidal neural network with polynomially bounded weights requires width exponential in $d$.

Our separation is robust and works for a general class of input distributions, as long as their density is at least $1/\text{poly}(d)$ on some small ball of radius $1/\text{poly}(d)$ in $\mathbb{B}^d$. The function we use can also be replaced by many other functions that are polynomially-Lipschitz but not close to any low-degree polynomial.

Our proof is simple and uses already known low-degree polynomial approximation to depth-2 neural networks that have been used in other contexts such as efficient learning algorithms for neural networks by Shalev-Shwartz et al. (2011), Livni et al. (2014), Goel & Klivans (2017). Our work builds upon Daniely (2017). Unlike that paper, while we have to place the extra restrictions of upper bounds on weights and type of nonlinearity allowed, our proof is much simpler than Danielys (for example, we dont use spherical harmonics) and this simplicity lends it flexibility allowing us to prove lower bounds for a general class of distributions.

## 3 $L_2$-SEPARATION OF DEPTH-2 VS. DEPTH-3 SEPARATION FOR GENERAL INPUT DISTRIBUTIONS

In this section, we show lower bounds under the $L_2$-norm. We say that a function $G : \mathbb{B}^d \to \mathbb{R}$ cannot be $\delta$-approximated by a class $\mathcal{F}$ of functions in $L_2$-norm for distribution $\mu$ on $\mathbb{B}^d$ if for all $F \in \mathcal{F}$ we have

$$\int_{\mathbb{B}^d} (G(\mathbf{x}) - F(\mathbf{x}))^2 \, \mu(\mathbf{x}) \, d\mathbf{x} \; > \; \delta.$$

In other words, this weighted squared error means that for every function $F$ in $\mathcal{F}$, function $G$ differs from $F$ in a significant portion of the domain as measured by $\mu$.

Theorem 1 below gives a technical condition on the class of densities $\mu$ on $\mathbb{B}^d$ for which our lower bound holds. Here is an example to illustrate that the condition on density is reasonable: Let $K \subset \mathbb{B}^d$ be any convex set such that $\mathbb{B}(\bar{0}, 1 - 2r) \subseteq K \subseteq \mathbb{B}(\bar{0}, 1 - r)$, where $r = 1/\text{poly}(d)$, and consider $\mu$ to be any probability density supported on $K$ that is within constant factors from the uniform density on $K$. The conditions required for the lower bound in Theorem 1 apply to this distribution. Note that Theorem 1 can tolerate zero probability mass over small subsets of $\mathbb{B}^d$, and works even with approximately uniform density at many points.

**Theorem 1.** *Consider the function $G : \mathbb{B}^d \to \mathbb{R}$ given by $G(\mathbf{x}) = \sin(\pi N \|\mathbf{x}\|^2)$. Let $\mu$ be any probability density over $\mathbb{B}^d$ such that there exists a subset $C \subseteq \mathbb{B}^d$ satisfying the following two conditions:*

- *The $r$-interior of $C$ defined as $C' = \{\mathbf{x} \in C : \mathbb{B}(\mathbf{x}, r) \subseteq C\}$ contains at least $\gamma$ fraction of the total probability mass for some $\gamma > 0$, i.e., $\int_{C'} \mu(\mathbf{x}) d\mathbf{x} \geq \gamma$.*

- *For any affine line $\ell$, the induced probability density on every segment of length at least $r$ in the intersection $\ell \cap C$ is $(\alpha, \beta)$-uniform, i.e., it is at least $\alpha$ times and at most $\beta$ times the uniform density on that segment.*

*Let $F : \mathbb{B}^d \to \mathbb{R}$ be any function computed by a depth-2 sigmoidal neural network with weights bounded by $B$ and width $n$. Then for any $0 < \delta \ll \alpha\gamma/3\beta$ and $N \gg (B/r^2) \log(nB^2/\delta)$, the function $F$ cannot $\delta$-approximate $G$ on $\mathbb{B}^d$ under $L_2$-norm (squared error) under the probability density $\mu$.*

*In particular, if $\alpha, \beta, \gamma$ are constants, $B = poly(d)$, $n = 2^d$, and $r = 1/poly(d)$, then it suffices to choose $N = poly(d)$ for a sufficiently large degree polynomial.*

*Remark.* Let us make some remarks on the weight restriction in our results as it is somewhat uncommon. Our depth separation results apply to neural networks with bounds on the magnitudes of the weights. While we would like to prove our results without any weight restrictions, our current proof technique needs this restriction. However, we now argue that small weights are also natural. In training neural networks, often weights are not allowed to be too large to avoid overfitting. Weight decay is a commonly used regularization heuristic in deep learning to control the weights. Early stopping can also achieve this effect. Another motivation to keep the weights low is to keep the Lipschitz constant of the function computed by the network (w.r.t. changes in the input, while keeping the network parameters fixed) small. Goodfellow et al. (2016) contains many of these references. One of the surprising discoveries about neural networks has been the existence of adversarial examples (Szegedy et al. (2013)). These are examples obtained by adding a tiny perturbation to input from class so that the resulting input is misclassified by the network. The perturbations are imperceptible to humans. Existence of such examples for a network suggests that the Lipschitz constant of the network is high as noted in Szegedy et al. (2013). This lead them to suggest regularizing training of neural nets by penalizing high Lipschitz constant to improve the generalization error and, in particular, eliminate adversarial examples. This is carried out in Cissé et al. (2017), who find a way to control the Lipschitz constant by enforcing an orthonormality constraint on the weight matrices along with other tricks. They report better resilience to adversarial examples. On the other hand, Neyshabur et al. (2017) suggest that Lipschitz constant cannot tell the full story about generalization.

*Remark.* The Lipschitz constant of our hard function is $\text{poly}(d)$, and it would be desirable to have smaller Lipschitz constant if possible. To our knowledge, all related previous work also has Lipschitz constant at least this large.

The $L_2$ separation between depth-2 and depth-3 neural networks under probability density $\mu$ now follows by taking a small enough $\delta$, and combining the following ingredients (i) Proposition 5 says that any depth-2 sigmoid neural networks of width $n = 2^d$ and weights bounded by $B = \text{poly}(d)$ can be $\delta$-approximated in $L_\infty$ (and hence, also $L_2$) by a multivariate polynomials of degree $D = O(B \log(nB^2/\delta)) = \text{poly}(d)$, (ii) proof of Theorem 8 (initial part) says that $G(\mathbf{x})$ can be $\delta$-approximated in $L_\infty$ (and hence, also $L_2$) by a depth-3 sigmoid neural network of width and size $\text{poly}(d)$, but (iii) Theorem 1 says that, for $N = \text{poly}(d)$ of large enough degree, $G(\mathbf{x})$ cannot be $3\delta$-approximated in $L_2$ by any multivariate polynomial of degree $D$, and (iv) triangle inequality.

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

## A    PROOF OF THEOREM 1

*Proof.* We show a lower bound on $L_2$-error of approximating $G(\mathbf{x})$ with any multivariate polynomial $P : \mathbb{B}^d \to \mathbb{R}$ of degree $D$ under the distribution given by $\mu$ on $\mathbb{B}^d$. For any fixed unit vector $\mathbf{v}$, consider $\mathbf{u} \in \mathbb{B}^{d-1}$ orthogonal to $\mathbf{v}$ and let $\ell_{\mathbf{u}}$ be the affine line going through $\mathbf{u}$ and parallel to the direction $\mathbf{v}$ given by $\ell_{\mathbf{u}} = \{\mathbf{x} = \mathbf{u} + t\mathbf{v} \ : \ t \in \mathbb{R}\}$.

$$\int\limits_{\mathbb{B}^d} \left(G(\mathbf{x}) - P(\mathbf{x})\right)^2 \mu(\mathbf{x})d\mathbf{x}$$

$$\geq \int\limits_{C} \left(G(\mathbf{x}) - P(\mathbf{x})\right)^2 \mu(\mathbf{x})d\mathbf{x}$$

$$= \int\limits_{\mathbb{B}^{d-1}} \int\limits_{\mathbb{R}} \left(G(\mathbf{u} + t\mathbf{v}) - P(\mathbf{u} + t\mathbf{v})\right)^2 \mu(\mathbf{u} + t\mathbf{v}) \, \mathbb{I}_C(\mathbf{u}, t) \, dt \, d\mathbf{u}$$

$$\text{where } \mathbb{I}_C(\mathbf{u}, t) = \begin{cases} 1, & \text{if } \mathbf{u} \in \mathbb{B}^d \text{ and } \mathbf{u} + t\mathbf{v} \in C \\ 0, & \text{otherwise} \end{cases}$$

$$\geq \int\limits_{\mathbb{B}^{d-1}} \int\limits_{\mathbb{R}} \left(G(\mathbf{u} + t\mathbf{v}) - P(\mathbf{u} + t\mathbf{v})\right)^2 \mu(\mathbf{u} + t\mathbf{v})\, \mathbb{I}_{\tilde{C}}(\mathbf{u}, t)\, dt\, d\mathbf{u}$$

where $\mathbb{I}_{\tilde{C}}(\mathbf{u}, t) = 1$, if $\mathbf{u} \in \mathbb{B}^{d-1}$ and $\mathbf{u} + t\mathbf{v} \in \{\mathbf{u} + t'\mathbf{v} \in C \ : \ t' \in I\} \subseteq C$
for some interval $I$ of length at least $r$, and $\mathbb{I}_{\tilde{C}}(\mathbf{u}, t) = 0$, otherwise

$$= \frac{\int_{\mathbb{B}^{d-1}} \int_{\mathbb{R}} \left(G(\mathbf{u} + t\mathbf{v}) - P(\mathbf{u} + t\mathbf{v})\right)^2 \mu(\mathbf{u} + t\mathbf{v})\, \mathbb{I}_{\tilde{C}}(\mathbf{u}, t)\, dt\, d\mathbf{u}}{\int_{\mathbb{B}^{d-1}} \int_{\mathbb{R}} \mu(\mathbf{u} + t\mathbf{v})\, \mathbb{I}_{\tilde{C}}(\mathbf{u}, t)\, dt\, d\mathbf{u}} \cdot \int\limits_{\mathbb{B}^{d-1}} \int\limits_{\mathbb{R}} \mu(\mathbf{u} + t\mathbf{v})\, \mathbb{I}_{\tilde{C}}(\mathbf{u}, t)\, dt\, d\mathbf{u}$$

$$\geq \frac{\int_{\mathbb{B}^{d-1}} \int_{\mathbb{R}} \left(G(\mathbf{u} + t\mathbf{v}) - P(\mathbf{u} + t\mathbf{v})\right)^2 \mu(\mathbf{u} + t\mathbf{v})\, \mathbb{I}_{\tilde{C}}(\mathbf{u}, t)\, dt\, d\mathbf{u}}{\int_{\mathbb{B}^{d-1}} \int_{\mathbb{R}} \mu(\mathbf{u} + t\mathbf{v})\, \mathbb{I}_{\tilde{C}}(\mathbf{u}, t)\, dt\, d\mathbf{u}} \cdot \int\limits_{C'} \mu(\mathbf{x}) d\mathbf{x}$$

because for any $\mathbf{x} = \mathbf{u} + t\mathbf{v} \in C'$ we have $\mathbb{B}(\mathbf{x}, r) \subseteq C$, therefore $\mathbb{I}_{\tilde{C}}(\mathbf{u}, t) = 1$

$$\geq \min_{\mathbf{u} \in \mathbb{B}^{d-1}} \frac{\int_{\mathbb{R}} \left(G(\mathbf{u} + t\mathbf{v}) - P(\mathbf{u} + t\mathbf{v})\right)^2 \mu(\mathbf{u} + t\mathbf{v})\, \mathbb{I}_{\tilde{C}}(\mathbf{u}, t)\, dt}{\int_{\mathbb{R}} \mu(\mathbf{u} + t\mathbf{v})\, \mathbb{I}_{\tilde{C}}(\mathbf{u}, t)\, dt} \cdot \int\limits_{C'} \mu(\mathbf{x}) d\mathbf{x}$$

$$\geq \min_{\mathbf{u} \in \mathbb{B}^{d-1}} \frac{\alpha}{\beta} \cdot \frac{\int_{\mathbb{R}} \left(G(\mathbf{u} + t\mathbf{v}) - P(\mathbf{u} + t\mathbf{v})\right)^2 \mathbb{I}_{\tilde{C}}(\mathbf{u}, t)\, dt}{\int_{\mathbb{R}} \mathbb{I}_{\tilde{C}}(\mathbf{u}, t)\, dt} \cdot \int\limits_{C'} \mu(\mathbf{x}) d\mathbf{x}$$

because for any line $\ell$, the distribution induced by $\mu(\mathbf{x})$ along any line segment
of length at least $r$ in the intersection $\ell \cap C$ is $(\alpha, \beta)$-uniform, for any line $\ell$

$$\geq \frac{\alpha}{\beta} \cdot \frac{\gamma}{3}.$$

The last inequality is using the condition $\int_{C'} \mu(\mathbf{x}) d\mathbf{x} \geq \gamma$ given in Theorem 1 and an adaptation of the following idea from Lemma 5 of Daniely (2017). For any fixed $\mathbf{u}$ and $\mathbf{v}$, $G(\mathbf{u} + t\mathbf{v}) = \sin(\pi N(\|\mathbf{u}\|^2 + t^2))$ and $P(\mathbf{u} + t\mathbf{v})$ is a polynomial of degree at most $D$ in $t$. The function $\sin(\pi N(\|\mathbf{u}\|^2 + t^2))$ alternates its sign as $\|\mathbf{u}\|^2 + t^2$ takes values that are successive integer multiples of $1/N$. Consider $s = t^2 \in [0, 1]$ and divide $[0, 1]$ into $N$ disjoint segments using integer grid of step size $1/N$. For any polynomial $p(s)$ of degree at most $D$ and any interval $I \subseteq [0, 1]$ of length $r \gg D/N$, there exists at least $Nr - D - 2$ segments of length $1/N$ each on which $\sin(\pi N s)$ and $p(s)$ do not change signs and have opposite signs. Now using $(\sin(\pi N s) - p(s))^2 \geq \sin^2(\pi N s)$, integrating we get that $\int_I (\sin(\pi N s) - p(s))^2 ds \geq r/2$. Extending this proof to $t$ instead of $s = t^2$, using $\sin^2(\pi N t^2) t \leq \sin^2(\pi N t)$ for all $t \in [0, 1]$, and incorporating the shift $\pi N \|\mathbf{u}\|^2$, we can similarly show that $\int_I \sin^2(\pi N(\|\mathbf{u}\|^2 + t^2)) - P(\mathbf{u} + t\mathbf{v}))^2 dt \geq r/3$. Summing up over multiple such intervals gives the final inequality. $\square$

## B  POLYNOMIAL APPROXIMATIONS TO SIGMOIDAL NEURAL NETWORKS

In this section, we show that a sigmoid neuron can be well-approximated by a low-degree polynomial. As a corollary, we show that depth-2 (and in genenral, small-depth) sigmoidal neural networks can be well-approximated by low-degree multivariate polynomials. The main idea is to use Chebyshev polynomial approximation as in Shalev-Shwartz et al. (2011), which closely approximates the minimax polynomial (or the polynomial that has the smallest maximum deviation) to a given function. For the simplicity of presentation and arguments, we drop the bias term $b$ in the activation function $\sigma(\langle \mathbf{w}, \mathbf{x} \rangle + b)$. This is without loss of generality, as explained at the end of the last section.

### B.1  POLYNOMIAL APPROXIMATION TO A SIGMOID NEURON

The activation function of a sigmoid neuron $\sigma : \mathbb{R} \to \mathbb{R}$ is defined as

$$\sigma(t) = \frac{1}{1 + \exp(-t)}.$$

Chebyshev polynomials of the first kind $\{T_j(t)\}_{j \geq 0}$ are defined recursively as $T_0(t) = 1$, $T_1(t) = t$, and $T_{j+1}(t) = 2t \cdot T_j(t) - T_{j-1}(t)$. They form an orthonormal basis of polynomials over $[-1, 1]$

with respect to the density $1/\sqrt{1-t^2}$. The coefficient $c_j$ in the Chebyshev expansion of $\sigma(wt) = \sum_{j=0}^{\infty} c_j T_j(t)$ over $[-1, 1]$ is given by

$$c_j = \frac{1 + \mathbf{1}(j > 0)}{\pi} \int_{-1}^{1} \frac{\sigma(wt) \, T_j(t)}{\sqrt{1-t^2}} dt.$$

Proposition 2 (see Lemma B.1 in Shalev-Shwartz et al. (2011)) bounds the magnitude of coefficients $c_j$ in the Chebyshev expansion of $\sigma(wt) = \sum_{j=0}^{\infty} c_j T_j(t)$.

**Proposition 2.** *For any $j > 1$, the coefficient $c_j$ in the Chebyshev expansion of a sigmoid neuron $\sigma(wt)$ is bounded by*

$$|c_j| \leq \left( \frac{4}{|w|} + \frac{2}{\pi} \right) \left( 1 + \frac{\pi}{|w|} \right)^{-j}.$$

Proposition 2 implies low-degree polynomial approximation to sigmoid neurons as follows. This observation appeared in Shalev-Shwartz et al. (2011) (see equation (B.7) in their paper). For completeness, we give the proof in Appendix D.

**Proposition 3.** *Given any $w \in \mathbb{R}$ with $|w| \leq B$, there exists a polynomial $p$ of degree $O\left(B \log\left(B/\epsilon\right)\right)$ such that $|\sigma(wt) - p(t)| \leq \epsilon$, for all $t \in [-1, 1]$.*

We use this $O\left(\log(1/\epsilon)\right)$ dependence in the above bound crucially in some of our results, e.g., a weaker version of Daniely's separation result for depth-2 and depth-3 neural networks. Notice that this logarithmic dependence does not hold for a ReLU neuron; it is $O(1/\epsilon)$ instead.

### B.2  POLYNOMIALS APPROXIMATIONS TO SMALL-DEPTH NEURAL NETWORKS

A depth-2 sigmoidal neural network on input $t \in [-1, 1]$ computes a linear combination of sigmoidal neurons $\sigma(w_1), \sigma(w_2 t), \ldots, \sigma(w_n t)$, for $w_1, w_2, \ldots, w_n \in \mathbb{R}$, and computes a function $f : [-1, 1] \to \mathbb{R}$ given by

$$f(t) = \sum_{i=1}^{n} a_i \sigma(w_i t)$$

Here are a few propositions on polynomial approximations to small-depth neural networks. For completeness, their proofs are included in Appendix D.

Proposition 4 shows that a depth-2 sigmoidal neural network of bounded weights and width is close to a low-degree polynomial. Building upon the ideas from Shalev-Shwartz et al. (2011), this observation appears as Theorem 4 in Livni et al. (2014)).

**Proposition 4.** *Let $f : [-1, 1] \to \mathbb{R}$ be a function computed by a depth-2 sigmoidal neural network of width $n$ and weights bounded by $B$. Then $f$ is $\delta$-approximated (in $L_\infty$-norm) over $[-1, 1]$ by a polynomial of degree $O\left(B \log\left(nB^2/\delta\right)\right)$.*

Now consider a depth-2 sigmoidal neural network on input $\mathbf{x} \in \mathbb{B}^d$, where $\mathbb{B}^d = \{\mathbf{x} \in \mathbb{R}^d : \|\mathbf{x}\| \leq 1\}$. It is given by a linear combination of sigmoidal activations applied to linear functions $\langle \mathbf{w}_1, \mathbf{x} \rangle, \langle \mathbf{w}_2, \mathbf{x} \rangle, \ldots, \langle \mathbf{w}_n, \mathbf{x} \rangle$ (or affine functions when we have biases), for $\mathbf{w}_1, \mathbf{w}_2, \ldots, \mathbf{w}_n \in \mathbb{R}^d$ and it computes a function $F : \mathbb{B}^d \to \mathbb{R}$ given by

$$F(\mathbf{x}) = \sum_{i=1}^{n} a_i \sigma(\langle \mathbf{w}_i, \mathbf{x} \rangle)$$

Proposition 5 below is a multivariate version of Proposition 4.

**Proposition 5.** *Let $F : \mathbb{B}^d \to \mathbb{R}$ be a function computed by a depth-2 sigmoidal neural network with width $n$ and bounded weights, that is, $|a_i| \leq B$ and $\|\mathbf{w}_i\| \leq B$, for $1 \leq i \leq n$. Then $F$ is $\delta$-approximated (in $L_\infty$-norm) over $\mathbb{B}^d$ by a polynomial of degree $O\left(B \log\left(nB^2/\delta\right)\right)$ in $d$ variables given by the coordinates $\mathbf{x} = (x_1, x_2, \ldots, x_d)$.*

Note that its proof crucially uses the fact that Proposition 3 guarantees a low-degree polynomial that approximates a sigmoid neuron everywhere in $[-1, 1]$.

A depth-$k$ sigmoidal neural network can be thought of as a composition – a depth-2 sigmoidal neural network on top, whose each input variable is a sigmoid applied to a depth-$(k-2)$ sigmoidal neural network. In other words, it computes a function $F : \mathbb{B}^d \to \mathbb{R}$ given by

$$F(\mathbf{x}) = \sum_{i=1}^{n} a_i \sigma\left(\langle \mathbf{w}_i, \mathbf{y} \rangle\right),$$

where $\mathbf{y} = (y_1, y_2, \ldots, y_m)$ has each coordinate $y_j = \sigma(F_j(\mathbf{x}))$, for $1 \leq j \leq m$, such that each $F_i : \mathbb{B}^d \to \mathbb{R}$ is a function computed by a depth-$(k-2)$ sigmoidal neural network.

Now we show an interesting consequence, namely, any constant-depth sigmoidal neural network with polynomial width and polynomially bounded weights can be well-approximated by a low-degree multivariate polynomial. The bounds presented in Proposition 6 are not optimal but the qualitative statement is interesting in contrast with the depth separation result. The growth of the degree of polynomial approximation is dependent on the widths of hidden layers and it is also the subtle reason why a depth separation result is still possible (when the weights are bounded).

**Proposition 6.** *Let $F : \mathbb{B}^d \to \mathbb{R}$ be a function computed by a depth-$k$ sigmoidal neural network of width at most $n$ in each layer and weights bounded by $B$, then $F(\mathbf{x})$ can be $\delta$-approximated (in $L_\infty$-norm) over $\mathbb{B}^d$ by a $d$-variate polynomial of degree $O\left((nB)^k \log^k (nB/\delta)\right)$ in each coordinate variable of $\mathbf{x} = (x_1, x_2, \ldots, x_d)$.*

*Note that when $n$ and $B$ are polynomial in $d$ and the depth $k$ is constant, then this low-degree polynomial approximation also has degree polynomial in $d$.*

## C  $L_\infty$-SEPARATION OF DEPTH-$2$ VS. DEPTH-$3$ SEPARATION FOR GENERAL INPUT DISTRIBUTIONS

In this section, we show Theorem 8 about separation in $L_\infty$-norm between depth-2 vs. depth-3 sigmoid neural networks. Although $L_\infty$-separation is weaker than $L_2$-separation, our $L_\infty$-separation result in Theorem 8 holds for a larger class of distributions compared to our $L_2$-separation result in Theorem 1. In fact, Theorem 8 works for any distribution over $\mathbb{B}^d$ whose support contains a ball of radius at least $1/\text{poly}(d)$ with probability density at least within $1/\text{poly}(d)$ of the uniform density everywhere inside it.

Daniely shows that if $g : [-1, 1] \to \mathbb{R}$ cannot be approximated by a polynomial of degree $O(d^2)$, then $G : \mathbb{S}^{d-1} \times \mathbb{S}^{d-1} \to \mathbb{R}$ defined as $G(\mathbf{x}, \mathbf{y}) = g(\langle \mathbf{x}, \mathbf{y} \rangle)$ cannot be approximated by any depth-2 neural network of polynomial size and (exponentially) bounded weights. Daniely shows this lower bound for a general neuron or activation function that includes sigmoids and ReLUs. Daniely then uses $G(\mathbf{x}, \mathbf{y}) = g(\langle \mathbf{x}, \mathbf{y} \rangle) = \sin(\pi d^3 \langle \mathbf{x}, \mathbf{y} \rangle)$ which, on the other hand, is approximable by a depth-3 ReLU neural network with polynomial size and polynomially bounded weights. This gives a separation between depth-2 and depth-3 ReLU neural networks w.r.t. uniform distribution over $\mathbb{S}^{d-1} \times \mathbb{S}^{d-1}$. Daniely's proof uses harmonic analysis on the unit sphere, and requires the uniform distribution on $\mathbb{S}^{d-1} \times \mathbb{S}^{d-1}$ in a crucial way.

We show a simple proof of separation between depth-2 and depth-3 sigmoidal neural networks that compute functions $F : \mathbb{B}^d \to \mathbb{R}$. Our proof works for a large class of distributions on $\mathbb{B}^d$ but requires the weights to be polynomially bounded.

The following lemma appears in Debao (1993). Assumption 1 in Eldan & Shamir (2016) and their version of this lemma for ReLU networks was used by Daniely (2017) in the proof of separation between the expressive power of depth-2 and depth-3 ReLU networks.

**Lemma 7.** *Let $f : [-1, 1] \to \mathbb{R}$ be any $L$-Lipschitz function. Then there exists a function $g : [-1, 1] \to \mathbb{R}$ computed by a depth-$2$ sigmoidal neural network such that*

$$g(t) = f(0) + \sum_{i=1}^{n} a_i \sigma(w_i t + b_i),$$

*the width $n$ as well as the weights are bounded by $\text{poly}(L, 1/\epsilon)$, and $|f(t) - g(t)| \leq \epsilon$, for all $t \in [-1, 1]$.*

Now we are ready to show the separation between depth-2 and depth-3 sigmoidal neural networks. The main idea, similar to Daniely (2017), is to exhibit a function that is Lipschitz but far from any low-degree polynomial. The Lipschitz property helps in showing that our function can be well-approximated by a depth-3 neural network of small size and small weights. However, being far from any low-degree polynomial, it cannot be approximated by any depth-2 neural network.

**Theorem 8.** *Consider the function $G : \mathbb{B}^d \to \mathbb{R}$ given by $G(\mathbf{x}) = \sin(\pi d^5 \|\mathbf{x}\|^2)$. Then $G$ can be $\delta$-approximated (in $L_\infty$-norm) by a depth-3 sigmoidal neural network of width and weights polynomially bounded in $d$. However, any function $F : \mathbb{B}^d \to \mathbb{R}$ computed by a depth-2 sigmoidal neural network with weights $O(d^2)$ cannot $\delta$-approximate $G$ even when its width $n$ is $2^{O(d)}$.*

*By modifying the function to $G(\mathbf{x}) = \sin(\pi N \|\mathbf{x}\|^2)$, this lower bound with $L_\infty$-norm holds for any distribution over $\mathbb{B}^d$ whose support contains a radial line segment of length at least $1/poly(d)$, by making $N = poly(d)$, for a large enough polynomial.*

*Remark: Given any distribution $\mu$ over $\mathbb{B}^d$ whose probability density is at least $1/poly(d)$ on some small ball of radius $1/poly(d)$, the lower bound or inapproximability by any depth-2 sigmoidal neural network can be made to work with $L_2$-norm (squared error), for a large enough $N = poly(d)$.*

*Proof.* First, we will show that $G(\mathbf{x})$ can be well-approximated by a depth-3 sigmoidal neural network of polynomial size and weights. The idea is similar to Daniely's construction for ReLU networks in Daniely (2017). By Lemma 7, there exists a function $f : [-1, 1] \to \mathbb{R}$ computed by a depth-2 sigmoidal neural network of size and weights bounded by $poly(d, 1/\epsilon)$ such that $\left|t^2 - f(t)\right| \leq \epsilon/10d^6$, for all $t \in [-1, 1]$. Thus, we can compute $x_i^2$ for each coordinate of $\mathbf{x}$ and add them up to get an $\epsilon$-approximation to $\|\mathbf{x}\|^2$ over $\mathbb{B}^d$. That is, there exists a function $S : \mathbb{B}^d \to \mathbb{R}$ computed by a depth-2 sigmoidal neural network of size and weights bounded by $poly(d, 1/\epsilon)$ such that $\left|S(x) - \|\mathbf{x}\|^2\right| \leq \epsilon/10d^5$, for all $\mathbf{x} \in \mathbb{B}^d$. Again, by Lemma 7, we can approximate $\sin(\pi d^3 t)$ over $[0, 1]$ using $f : [-1, 1] \to \mathbb{R}$ computed by another depth-2 sigmoidal neural network with size and weights bounded by $poly(d, 1/\epsilon)$ such that $\left|\sin(\pi d^3 t) - f(t)\right| \leq \epsilon/2$, for all $t \in [0, 1]$. Note that the composition of these two depth-2 neural networks $f(N(\mathbf{x}))$ gives a depth-3 neural network as the output of the hidden layer of the bottom network can be fed into the top network as inputs.

$$
\begin{aligned}
|G(\mathbf{x}) - f(S(\mathbf{x}))| &= \left|\sin(\pi d^5 \|\mathbf{x}\|^2) - f(S(\mathbf{x}))\right| \\
&\leq \left|\sin(\pi d^5 \|\mathbf{x}\|^2) - f(\|\mathbf{x}\|^2)\right| + \left|f(\|\mathbf{x}\|^2) - f(S(\mathbf{x}))\right| \\
&\leq \epsilon/2 + 4d^5 \left|\|\mathbf{x}\|^2 - S(\mathbf{x})\right|
\end{aligned}
$$

using that $f$ that approximates $\sin(\pi d^5 t)$ closely must also be $4d^5$-Lipschitz

$$
\leq \epsilon/2 + 4d^5 \cdot \epsilon/10d^5 \leq \epsilon.
$$

Now we will show the lower bound. Consider any function $F : \mathbb{B}^d \to \mathbb{R}$ computed by a depth-2 sigmoidal neural network whose weights are bounded by $B = O(d^2)$ and width is $n$. Proposition 5 shows that there exists a $d$-variate polynomial $P(\mathbf{x})$ of degree $O\left(B \log(nB^2/\delta)\right) = O\left(d^2 \log(n/\delta) + d^2 \log d\right)$ in each variable such that $|F(\mathbf{x}) - P(\mathbf{x})| \leq \delta$, for all $\mathbf{x} \in \mathbb{B}^d$. Let $\mu$ be any measure on $\mathbb{B}^d$ whose support contains some radial line segment of length at least $1/poly(d)$ in $\mathbb{B}^d$. In other words, there exists a unit vector $\mathbf{u}$ such that the support of $\mu$ intersects the radial set $\{\mathbf{x} \in \mathbb{B}^d : \mathbf{x} = t\mathbf{u}, \text{for some } t \in [-1, 1]\}$ in some line segment of length at least $1/poly(d)$. Then $P(t\mathbf{u})$ is a univariate polynomial of degree $O(d^3 \log(n/\delta) + d^3 \log d)$ that $\delta$-approximates $F(t\mathbf{u})$, for all $t \in [-1, 1]$. By Lemma 9, using $D = O(d^3 \log(n/\delta) + d^3 \log d)$, $l = 1/poly(d)$ and $N = d^5/l$, we get that if $n = 2^{O(d)}$, then there exists a $t_0 \in [-1, 1]$ such that $\left|\sin(\pi N t_0^2) - P(t\mathbf{u})\right| \geq 1$. Therefore, by triangle inequality, $\left|\sin(\pi N \|t_0\mathbf{u}\|^2) - F(t_0\mathbf{u})\right| \geq \left|\sin(\pi N t_0^2) - P(t_0\mathbf{u})\right| - |P(t_0\mathbf{u}) - F(t_0\mathbf{u})| \geq 1 - \delta > \delta$, for $\delta < 1/2$. This means that $G(\mathbf{x})$ cannot be well-approximated by any $F(\mathbf{x})$ computed by a depth-2 neural network with polynomially bounded weights even when it has width $2^{O(d)}$. $\qquad \square$

Now we show that the candidate function proposed by Daniely $g(t) = \sin(\pi N t)$, for large enough $N$, is far from any low-degree polynomial w.r.t. any measure $\mu$ on $[-1, 1]$ with a reasonable support.

**Lemma 9.** *Let $p$ be any polynomial of degree $D$ and $\mu$ be any measure on $[-1, 1]$ whose support contains an interval of length at least $l$. Then, for $N$ large enough to satisfy $Nl > D + 3$, there exists $t_0 \in [-1, 1]$ such that $\mu(t_0) > 0$ and $|\sin(\pi N t_0) - p(t_0)| > 1$. In other words, $\sin(\pi N t)$ is 1-far (in $L_\infty$-norm) from any polynomial of degree $D$ over interval $[-1, 1]$ with measure $\mu$.*

*Proof.* Let $\mu(t) > 0$ for some interval $[a, a + l] \subseteq [-1, 1]$. Consider $S = \{t \in [a, a+l] \; : \; t = -1 + (i + 1/2)/N, \text{ for some integer } i\}$. Then $S$ contains at least $Nl - 2$ points where $\sin(\pi N t)$ alternates as $\pm 1$. Any polynomial $p$ of degree $D$ cannot match the sign of $\sin(\pi N t)$ on all the points in $S$. Otherwise, by intermediate value theorem, $p$ must have at least $Nl - 3$ roots between the points of $S$, which means $D \geq Nl - 3$, a contradiction. Thus, there exists $t_0 \in S$ such that $p(t_0)$ and $\sin(\pi N t_0)$ have opposite signs. Since $\sin(\pi N t) = \pm 1$, for any $t \in S$, the sign mismatch implies $|\sin(\pi N t_0) - p(t_0)| > 1$. □

*An important remark on biases:* Even though we handled the case of sigmoid neurons without biases, the proof technique carries over to the sigmoid neurons with biases $\sigma(\langle \mathbf{w}, \mathbf{x} \rangle + b)$. The idea is to consider a new $(d + 1)$-dimensional input $\mathbf{x}_{\text{new}} = (\mathbf{x}, x_{d+1}) = (x_1, x_2, \ldots, x_{d+1})$ with $x_{d+1} = 1$, and consider the new weight vector $\mathbf{w}_{\text{new}} = (\mathbf{w}, b)$. Thus, $\langle \mathbf{w}_{\text{new}}, \mathbf{x}_{\text{new}} \rangle = \langle \mathbf{w}, \mathbf{x} \rangle + b$. The new input lies on a $d$-dimensional hyperplane slice of $\mathbb{B}^{d+1}$, so we need to look at the restriction of the input distribution $\mu$ to this slice. Most of the ideas in our proofs generalize without any technical modifications. We defer the details to the full version.

## D    PROOFS OF POLYNOMIAL APPROXIMATIONS TO NEURAL NETWORKS

**Proof of Proposition 3**

*Proof.* Consider the degree-$D$ approximation to $\sigma(wt)$ given by the first $D$ terms in its Chebyshev expansion. The error of this approximation for any $t \in [-1, 1]$ is bounded by

$$
\begin{aligned}
|\sigma(wt) - p(t)| &= \left| \sum_{j > D} c_j T_j(t) \right| \\
&\leq \sum_{j > D} |c_j| \\
&\leq \left( \frac{4}{|w|} + \frac{2}{\pi} \right) \sum_{j > D} \left( 1 + \frac{\pi}{|w|} \right)^{-j} \\
&\leq \left( \frac{4}{|w|} + \frac{2}{\pi} \right) \left( 1 + \frac{\pi}{|w|} \right)^{-(D+1)} \sum_{j=0}^{\infty} \left( 1 + \frac{\pi}{|w|} \right)^{-j} \\
&= \left( \frac{4}{|w|} + \frac{2}{\pi} \right) \left( 1 + \frac{\pi}{|w|} \right)^{-(D+1)} \cdot \frac{|w|}{\pi} \left( 1 + \frac{\pi}{|w|} \right) \\
&\leq \epsilon,
\end{aligned}
$$

using Proposition 2, $|w| \leq B$, and $D = O(B \log(B/\epsilon))$. □

**Proof of Proposition 4**

*Proof.* Let $f$ be computed by a depth-2 sigmoidal neural network given by $f(t) = \sum_{i=1}^{n} a_i \sigma(w_i t)$. Define a parameter $\epsilon = \delta/nB$. Proposition 3 guarantees polynomial $p_1, p_2, \ldots, p_n$ of degree $O(B \log(B/\epsilon))$ such that $|\sigma(w_i t) - p_i(t)| \leq \epsilon$, for all $t \in [-1, 1]$. Thus, the polynomial $p(t) = \sum_{i=1}^{n} a_i p_i(t)$ has degree $O(B \log(B/\epsilon)) = O(B \log(nB^2/\delta))$, and for any $t \in [-1, 1]$,

$$
\left| \sum_{i=1}^{n} a_i \sigma(w_i t) - p(t) \right| = \left| \sum_{i=1}^{n} a_i \sigma(w_i t) - \sum_{i=1}^{n} a_i p_i(t) \right|
$$

$$\leq \sum_{i=1}^{n} |a_i| \, |\sigma(w_i t) - p_i(t)|$$
$$\leq nB\epsilon \qquad\qquad\qquad\qquad \text{using } |a_i| \leq B$$
$$= \delta \qquad\qquad\qquad\qquad\qquad \text{using } \epsilon = \delta/nB.$$

$\square$

## Proof of Proposition 5

*Proof.* Let $F$ be computed by a depth-2 neural network given by $F(\mathbf{x}) = \sum_{i=1}^{n} a_i \sigma(\langle \mathbf{w}_i, \mathbf{x} \rangle)$, where $|a_i| \leq B$ and $\|\mathbf{w}_i\| \leq B$, for $1 \leq i \leq n$. Thus, $F(\mathbf{x}) = \sum_{i=1}^{n} a_i \sigma(\|\mathbf{w}_i\| \, t_i)$, where $t_i = \langle \mathbf{w}_i / \|\mathbf{w}_i\|, \mathbf{x} \rangle \in [-1, 1]$ because $\mathbf{w}_i / \|\mathbf{w}_i\| \in \mathbb{B}^d$, for $1 \leq i \leq n$, and $\mathbf{x} \in \mathbb{B}^d$.

Define a parameter $\epsilon = \delta/nB$. Proposition 3 guarantees polynomial $p_1, p_2, \ldots, p_n$ of degree $O\left(B \log(B/\epsilon)\right)$ such that $|\sigma(\|w_i\| \, t) - p_i(t)| \leq \epsilon$, for all $t \in [-1, 1]$. Consider the following polynomial $P(\mathbf{x}) = P(x_1, x_2, \ldots, x_d) = \sum_{i=1}^{n} a_i p_i(\langle \mathbf{w}_i / \|\mathbf{w}_i\|, \mathbf{x} \rangle)$. $P(\mathbf{x})$ is a $d$-variate polynomial of degree $O\left(B \log(B/\epsilon)\right) = O\left(B \log(nB^2/\delta)\right)$ in each variable $x_1, x_2, \ldots, x_d$. For any $\mathbf{x} \in \mathbb{B}^d$,

$$|F(\mathbf{x}) - P(\mathbf{x})| = \left| \sum_{i=1}^{n} a_i \sigma(\langle \mathbf{w}_i, \mathbf{x} \rangle) - \sum_{i=1}^{n} a_i p_i(\langle \mathbf{w}_i / \|\mathbf{w}_i\|, \mathbf{x} \rangle) \right|$$
$$\leq \sum_{i=1}^{n} |a_i| \, |\sigma(\langle \mathbf{w}_i, \mathbf{x} \rangle) - p_i(\langle \mathbf{w}_i / \|\mathbf{w}_i\|, \mathbf{x} \rangle)|$$
$$\leq B \sum_{i=1}^{n} |\sigma(\|\mathbf{w}_i\| \, t_i) - p_i(t_i)| \qquad \text{using } t_i = \langle \mathbf{w}_i / \|\mathbf{w}_i\|, \mathbf{x} \rangle \text{ and } |a_i| \leq B$$
$$\leq \epsilon nB \qquad \text{using } |\sigma(\|w_i\| \, t) - p_i(t)| \leq \epsilon, \text{ for all } t \in [-1, 1]$$
$$= \delta.$$

$\square$

## Proof of Proposition 6

*Proof.* We prove this by induction on the depth $k$. By induction hypothesis each $F_j(\mathbf{x})$ can be $\epsilon_1$-approximated (in $L_\infty$-norm) by a $d$-variate polynomial $Q_j(\mathbf{x})$ of degree $O\left((nB)^{k-2} \log^{(k-2)}(nB/\epsilon_1)\right)$ in each variable. Thus, $|F_j(\mathbf{x}) - Q_j(\mathbf{x})| = \epsilon_1$, for any $\mathbf{x} \in \mathbb{B}^d$ and $1 \leq j \leq m$. Because a sigmoid neuron is Lipschitz,

$$|y_j - \sigma(Q_j(\mathbf{x}))| = |\sigma(F_j(\mathbf{x})) - \sigma(Q_j(\mathbf{x}))| \leq |F_j(\mathbf{x}) - Q_j(\mathbf{x})| \leq \epsilon_1,$$

for any $\mathbf{x} \in \mathbb{B}^d$ and $1 \leq j \leq m$.

Since $F_j(\mathbf{x})$ is the output of a depth-$(k-2)$ sigmoidal neural network of width at most $n$ and weights at most $B$, we must have $|F_j(\mathbf{x})| \leq nB$, for all $\mathbf{x} \in \mathbb{B}^d$. Thus, $|Q_j(\mathbf{x})| \leq nB + \epsilon_1 \leq 2nB$. By Proposition 3, there exists a polynomial $q(t)$ of degree at most $O\left(nB \log(nB/\epsilon_2)\right)$ such that

$$|\sigma(Q_j(\mathbf{x})) - q(Q_j(\mathbf{x}))| \leq \epsilon_2,$$

for all $\mathbf{x} \in \mathbb{B}^d$ and $1 \leq j \leq m$.

Consider $\mathbf{q} \in \mathbb{R}^m$ as $\mathbf{q} = (q(Q_1(\mathbf{x})), q(Q_2(\mathbf{x})), \ldots, q(Q_m(\mathbf{x})))$. Then, for any $\mathbf{x} \in \mathbb{B}^d$, we have

$$|\langle \mathbf{w}_i, \mathbf{y} \rangle - \langle \mathbf{w}_i, \mathbf{q} \rangle| = |\langle \mathbf{w}_i, \mathbf{y} - \mathbf{q} \rangle|$$
$$\leq \|\mathbf{w}_i\| \, \|\mathbf{y} - \mathbf{q}\|$$
$$\leq B \left( \sum_{j=1}^{m} (y_j - q(Q_j(\mathbf{x})))^2 \right)^{1/2}$$

$$\leq B\sqrt{m}\left(\epsilon_1 + \epsilon_2\right)$$
$$\leq B\sqrt{n}\left(\epsilon_1 + \epsilon_2\right).$$

Again by Proposition 3, there is a polynomial $p$ of degree at most $O\left(nB\log(nB/\epsilon)\right)$ such that $|\sigma(\langle \mathbf{w}_i, \mathbf{q}\rangle) - p(\langle \mathbf{w}_i, \mathbf{q}\rangle)| \leq \epsilon$, for all $\mathbf{x} \in \mathbb{B}^d$ and $1 \leq i \leq n$. This is because $|\langle \mathbf{w}_i, \mathbf{q}\rangle| = O(nB)$.

Let's define $P(\mathbf{x}) = \sum_{i=1}^{n} a_i p(\langle \mathbf{w}_i, \mathbf{q}\rangle)$. Therefore, for any $\mathbf{x} \in \mathbb{B}^d$,

$$
\begin{aligned}
|F(\mathbf{x}) - P(\mathbf{x})| &= \left| \sum_{i=1}^{n} a_i \sigma(\langle \mathbf{w}_i, \mathbf{y}\rangle) - \sum_{i=1}^{n} a_i p(\langle \mathbf{w}_i, \mathbf{q}\rangle) \right| \\
&\leq \sum_{i=1}^{n} |a_i| \, |\sigma(\langle \mathbf{w}_i, \mathbf{y}\rangle) - p(\langle \mathbf{w}_i, \mathbf{q}\rangle)| \\
&\leq \sum_{i=1}^{n} |a_i| \left( |\sigma(\langle \mathbf{w}_i, \mathbf{y}\rangle) - \sigma(\langle \mathbf{w}_i, \mathbf{q}\rangle)| + |\sigma(\langle \mathbf{w}_i, \mathbf{q}\rangle) - p(\langle \mathbf{w}_i, \mathbf{q}\rangle)| \right) \\
&\leq \sum_{i=1}^{n} |a_i| \left( |\langle \mathbf{w}_i, \mathbf{y}\rangle - \langle \mathbf{w}_i, \mathbf{q}\rangle| + \epsilon \right) \\
&\leq nB \left( B\sqrt{n}(\epsilon_1 + \epsilon_2) + \epsilon \right) \\
&\leq \delta,
\end{aligned}
$$

if we use $\epsilon_1 = \epsilon_2 = \delta/3n^{3/2}B^2$ and $\epsilon = \delta/3nB$.

$P(\mathbf{x})$ is a $d$-variate polynomial of degree

$$\deg(P) \leq \deg(p)\deg(q) \cdot \deg(Q_j) = O\left( (nB)^k \log^k (nB/\delta) \right),$$

in each variable. $\qquad\square$

## E   OTHER RELATED WORK

Telgarsky (2016) shows a separation between depth-$2k^3 + 8$ and depth-$k$ ReLU neural networks, for any positive integer $k$, when the input is uniformly distributed over $[-1, 1]^d$. Liang & Srikant (2017) (see also Safran & Shamir (2017); Yarotsky (2016)) show that there are univariate functions on a bounded interval such that neural networks of constant depth require size at least $\Omega\left(\text{poly}(1/\epsilon)\right)$ for a uniform $\epsilon$-approximation over the interval, whereas deep networks (the depth can depend on $\epsilon$) can have size $O\left(\text{polylog}(1/\epsilon)\right)$.

Shamir (2016); Shalev-Shwartz et al. (2017); Song et al. (2017) show that even functions computed by a depth-2 neural network of polynomial size can be hard to learn using gradient descent-type algorithms for a wide class of distributions. These results address questions about learnability rather than the expressive power of deep neural networks. Andoni et al. (2014) study the question of learning a low-degree polynomial using a randomly initialized depth-2 neural network and gradient descent. Goel & Klivans (2017) give a polynomial time algorithm for learning neural networks with one hidden layer of sigmoids feeding into any smooth, monotone activation function (e.g., ReLU, sigmoid). Both these results, especially Goel & Klivans (2017), show low-degree polynomial approximations to depth-2 sigmoid neural networks in their proof.

Hanin (2017) shows that piecewise affine functions on $[0, 1]^d$ with $N$ pieces can be exactly represented by a width-$(d + 3)$ network of depth at most $N$. Lower bound of $\Omega((N + d - 1)/(d + 1))$ on the depth is proven for functions of the above type when the network has width at most $(d + 1)$ and very closely approximates the function.

