# OpenReview forum: "Depth separation and weight-width trade-offs for sigmoidal neural networks"
_ICLR.cc/2018/Workshop — Accept_

### Official Review · AnonReviewer3 · 2018-03-05
**A simple example of separation**

**Rating:** 8
**Confidence:** 3

**Review:**

This paper provides an example of a function which can not be approximated with a depth 2 neural network (unless there are exponentially many hidden units) but can be approximated with a depth three neural network. While there are a few recent papers that do this. The main novelty of this paper is to study the effect of input distributions and show this separation holds for a variety of such distributions. Overall this seems like a short interesting example/paper. While I did not check proofs in detail the overall strategy seems sound.


- The conditions on the distributions are a bit unintuitive. More descriptions examples and why these conditions are relevant would be useful.

- There needs to be more discussion why the stated theorem achieves the objective claimed earlier on in the intro.

---

### Official Review · AnonReviewer2 · 2018-03-11
**Simpler proof of depth separation, quite robust to the underlying distribution**

**Rating:** 8
**Confidence:** 4

**Review:**

Summary: In this paper, authors show a depth separation between width 2 and width 3 sigmoidal neural networks. While there have been several results of such flavor in the recent literature, all of these results are strongly tied to the underlying distributions. In particular, it’s not clear from the previous results if the depth separation is robust under a larger class of distributions.
In this paper, the authors build on the work of Daniely(2016), giving a much simpler proof of his depth separation (though with worse parameters and a requirement of bounded weights), but the results in this paper hold for a much larger class of distributions.

Opinion: I think a simpler proof of depth separation and robustness to the underlying input distribution make the results, though weaker and not entirely new, interesting enough to merit acceptance.

---

### Decision · Program_Chairs · 2018-03-20
**ICLR 2018 Workshop Acceptance Decision**

**Decision:**

Accept

**Comment:**

Congratulations, your paper was accepted to the ICLR workshop.